# Robust Tensor Decomposition with Gross Corruption

**Quanquan Gu**[*]
Department of Operations Research
and Financial Engineering
Princeton University
Princeton, NJ 08544
`qgu@princeton.edu`

**Huan Gui**[*]  **Jiawei Han**
Department of Computer Science
University of Illinois
at Urbana-Champaign
Urbana, IL 61801
`{huangui2,hanj}@illinois.edu`

## Abstract

In this paper, we study the statistical performance of robust tensor decomposition with gross corruption. The observations are noisy realization of the superposition of a low-rank tensor $\mathcal{W}^*$ and an entrywise sparse corruption tensor $\mathcal{V}^*$. Unlike conventional noise with bounded variance in previous convex tensor decomposition analysis, the magnitude of the gross corruption can be arbitrary large. We show that under certain conditions, the true low-rank tensor as well as the sparse corruption tensor can be recovered simultaneously. Our theory yields nonasymptotic Frobenius-norm estimation error bounds for each tensor separately. We show through numerical experiments that our theory can precisely predict the scaling behavior in practice.

## 1 Introduction

Tensor data analysis have witnessed increasing applications in machine learning, data mining and computer vision. For example, an ensemble of face images can be modeled as a tensor, whose mode corresponds to pixels, subjects, illumination and viewpoint [23]. Traditional tensor decomposition methods such as Tucker decomposition and CANDECOMP/PARAFAC(CP) decomposition [14, 13] aim to factorize an input tensor into a number of low-rank factors. However, they are prone to local optima because they are solving essentially non-convex optimization problems. In order to address this problem, [15] [20] extended the trace norm of matrices [19] to tensors, and generalized convex matrix completion [8] [7] and matrix decomposition [6] to convex tensor completion/decomposition. For example, the goal of tensor decomposition aims to accurately estimate a low-rank tensor $\mathcal{W} \in \mathbb{R}^{n_1 \times \cdots \times n_K}$ from the noisy observation tensor $\mathcal{Y} \in \mathbb{R}^{n_1 \times \cdots \times n_K}$ that is contaminated by dense noises, i.e., $\mathcal{Y} = \mathcal{W}^* + \mathcal{E}$, where $\mathcal{W}^* \in \mathbb{R}^{n_1 \times \cdots \times n_K}$ is a low-rank tensor, $\mathcal{E} \in \mathbb{R}^{n_1 \times \cdots \times n_K}$ is a noise tensor whose entries are i.i.d. Gaussian noise with zero mean and bounded variance $\sigma^2$, i.e., $\mathcal{E}_{i_1,\ldots,i_K} \sim N(0, \sigma^2)$. [22] [21] analyzed the statistical performance of convex tensor decomposition under different extensions of trace norm. They showed that, under certain conditions, the estimation error scales with the rank of the true tensor $\mathcal{W}^*$. Furthermore, they demonstrated that given a noisy tensor, the true low-rank tensor can be recovered under restricted strong convexity assumption [18]. However, all these algorithms [15] [20] and theoretical results [22] [21] reply on the assumption that the observation noise has a bounded variance $\sigma^2$. Without this assumption, we are not able to identify the rank of $\mathcal{W}^*$, and therefore the estimated low-rank tensor $\widehat{\mathcal{W}}$ could be very far from the true tensor $\mathcal{W}^*$.

On the other hand, in many practical applications such as face recognition and image/video denoising, a portion of the observation tensor $\mathcal{Y}$ might be contaminated by gross error due to illumination, occlusion or pepper/salt noise. This scenario is not covered by finite variance noise assumption, therefore new mathematical models are demanded to address this problem. This motivates us to study

---

[*]Equal Contribution

convex tensor decomposition with gross corruption. It is clear that if all the entries of a tensor are corrupted by large error, there is no hope to recover the underlying low-rank tensor. To overcome this problem, one common assumption is that the gross corruption is sparse. Under this assumption, together with previous low-rank assumption, we formalize the noisy linear observation model as follows:

$$\mathcal{Y} = \mathcal{W}^* + \mathcal{V}^* + \mathcal{E}, \tag{1}$$

where $\mathcal{W}^* \in \mathbb{R}^{n_1 \times \dots \times n_K}$ is a low-rank tensor, $\mathcal{V}^* \in \mathbb{R}^{n_1 \times \dots \times n_K}$ is a sparse corruption tensor, where the locations of nonzero entries are unknown and the magnitudes of the nonzero entries can be arbitrarily large, and $\mathcal{E} \in \mathbb{R}^{n_1 \times \dots \times n_K}$ is a noise tensor whose entries are i.i.d. Gaussian noise with zero mean and bounded variance $\sigma^2$, and thus dense. Our goal is to recover the low-rank tensor $\mathcal{W}^*$, as well as the sparse corruption tensor $\mathcal{V}^*$. Note that in some applications, the corruption tensor is of independent interest and needs to be recovered.

Given the observation model in (1), and the low-rank as well as sparse assumptions on $\mathcal{W}^*$ and $\mathcal{E}^*$ respectively, we propose the following convex minimization to estimate the unknown low-rank tensor $\mathcal{W}^*$ and the sparse corruption tensor $\mathcal{E}^*$ simultaneously:

$$\arg\min_{\mathcal{W},\mathcal{V}} \|\|\mathcal{Y} - \mathcal{W} - \mathcal{V}\|\|_F^2 + \lambda_M \|\|\mathcal{W}\|\|_{S_1} + \mu_M \|\|\mathcal{V}\|\|_1, \tag{2}$$

where $\|\|\cdot\|\|_{S_1}$ is tensor Schatten-1 norm [22], $\|\|\cdot\|\|_1$ is entry-wise $\ell_1$ norm of tensors, and $\lambda_M$ and $\mu_M$ are positive regularization parameters. We call this optimization *Robust Tensor Decomposition*, which can been seen as a generalization of convex tensor decomposition in [15] [20] [22]. The regularization associated with the $\mathcal{E}$ encourages sparsity on the corruption tensor, where parameter $\mu_M$ controls the sparsity level. In this paper, we focus on the following questions: under what conditions for the size of the tensor, the rank of the tensor, and the fraction (sparsity level) of the corruption so that: (i) (2) is able to recover $\mathcal{W}^*$ and $\mathcal{V}^*$ with small estimator error? (ii) (2) is able to recover the exact rank of $\mathcal{W}^*$ and the support of $\mathcal{V}^*$? We will present nonasymptotic error bounds to answer these questions. Experiments on synthetic datasets validate our theoretical results.

The rest of this paper is arranged as follows. Related work is discussed in Section 2. Section 3 introduces the background and notations. Section 4 presents the main results. Section 5 provides an ADMM algorithm to solve the problem, followed by the numerical experiments in Section 6. Section 7 concludes this work with remarks.

## 2  Related Work

The problem of recovering the data under gross error has gained many attentions recently in matrix decomposition. A large body of work have been proposed and analyzed statistically. For example, [9] considered the problem of recovering an unknown low-rank and an unknown sparse matrix, given the sum of the two matrices. [5] proposed a similar problem, namely robust principal component analysis (RPCA), which studies the problem of recovering the low-rank and sparse matrices by solving a convex program. [10] studied multi-task regression which decomposes the coefficient matrix into two matrices, and imposes different group sparse regularization on two matrices. [25] considered more general case, where the parameter matrix could be the superposition of more than two matrices with different structurally constraints. Our paper extends [5] from two perspective: we extend the problem from matrices to high-order tensors, and consider the additional noise setting. We notice that [16] extended RPCA to tensors, which aims to recover the low-rank and sparse tensors by solving a constrained convex program. However, our formulation departs from [16] in that we consider not only the sparse corruption, but also the dense noise. We also note that low-rank noisy matrix completion [17] and robust matrix decomposition [1] [12] have been studied in in the high dimensional setting as well. Our model can be seen as the high-order extension of robust matrix decomposition. This extension is nontrivial, because the treatment of the tensor trace norm (Schatten-1 norm) is more complicated. More importantly, for the robust matrix decomposition problem considered [1], only the sum of error bound of two matrices (low-rank matrix and the sparse corruption matrix) can be obtained under the assumption of restricted strongly convexity. In contrast, under a different condition, our analysis provides error bound for each tensor component (low-rank tensor and the sparse corruption tensor) separately, making our results more appealing in practice and of independent theoretical interest. Since the problem in [1] is a special case of our problem, our

technical tool can be directly applied to their problem and yields new error bounds on the low-rank matrix as well as the sparse corruption matrix separately.

## 3 Notation and Background

Before proceeding, we define our notation and state assumptions that will appear in various parts of the analysis. For more details about tensor algebra, please refer to [14].

Scalars are denoted by lower case letters $(a, b, \ldots)$, vectors by bold lower case letters $(\boldsymbol{a}, \boldsymbol{b}, \ldots)$, matrices by bold upper case letters $(\mathbf{A}, \mathbf{B}, \ldots)$, and high-order tensors by calligraphic upper case letters $(\mathcal{A}, \mathcal{B}, \ldots)$. A tensor is a higher order generalization of a vector (first order tensor) and a matrix (second order tensor). From a multi-linear algebra view, tensor is a multi-linear mapping over a set of vector spaces. The order of tensor $\mathcal{A} \in \mathbb{R}^{n_1 \times \ldots \times n_2 \times \ldots \times n_K}$ is $K$, where $n_k$ is the dimensionality of the $k$-th order. Elements of $\mathcal{A}$ are denoted as $\mathcal{A}_{i_1 \ldots i_k \ldots i_n}, 1 \leq i_k \leq n_k$. We denote the number of elements in $\mathcal{A}$ by $N = \prod_{k=1}^{K} n_k$.

The mode-$k$ vectors of a $K$ order tensor $\mathcal{A}$ are the $n_k$ dimensional vectors obtained from $\mathcal{A}$ by varying index $i_k$ while keeping the other indices fixed. The mode-$k$ vectors are the column vectors of mode-$k$ flattening matrix $\mathbf{A}_{(k)} \in \mathbb{R}^{n_k \times (n_1 \ldots n_{k-1} n_{k+1} \ldots n_K)}$ that results by mode-$k$ flattening the tensor $\mathcal{A}$. For example, matrix column vectors are referred to as mode-1 vectors and matrix row vectors are referred to as mode-2 vectors.

The scalar product of two tensors $\mathcal{A}, \mathcal{B} \in \mathbb{R}^{n_1 \ldots n_2 \ldots n_K}$, is defined as $\langle \mathcal{A}, \mathcal{B} \rangle = \sum_{i_1} \ldots \sum_{i_K} \mathcal{A}_{i_1 \ldots i_K} \mathcal{B}_{i_1 \ldots i_K} = \mathrm{vec}(\mathcal{A}) \mathrm{vec}(\mathcal{B})$, where $\mathrm{vec}(\cdot)$ is a vectorization. The Frobenius norm of a tensor $\mathcal{A}$ is $\|\mathcal{A}\|_F = \sqrt{\langle \mathcal{A}, \mathcal{A} \rangle}$.

There are multiple ways to define tensor rank. In this paper, following [22], we define the rank of a tensor based on the mode-$k$ rank of a tensor. More specifically, the mode-$k$ rank of a tensor $\mathcal{X}$, denoted by $\mathrm{rank}_k(X)$, is the rank of the mode-$k$ unfolding $\mathbf{X}_{(k)}$ (note that $\mathbf{X}_{(k)}$ is a matrix, so its rank is well-defined). Based on mode-$k$ rank, we define the rank of tensor $\mathcal{X}$ as $r(\mathcal{X}) = (r_1, \ldots, r_k)$ if the mode-$k$ rank is $r_k$ for $k = 1, \ldots, K$. Note that the mode-$k$ rank can be computed in polynomial time, because it boils down to computing a matrix rank, whereas computing tensor rank [14] is NP complete.

Similarly, we extend the trace norm (a.k.a. nuclear norm) of matrices [19] to tensors. The overlapped Schatten-1 norm is defined as $\|\mathcal{X}\|_{S_1} = \frac{1}{K} \sum_{k=1}^{K} \|\mathbf{X}_{(k)}\|_{S_1}$, where $\mathbf{X}_{(k)}$ is the mode-$k$ unfolding of $\mathcal{X}$, and $\|\cdot\|_{S_1}$ is the Schatten-1 norm for a matrix, $\|\mathbf{X}\|_{S_1} = \sum_{j=1}^{r} \sigma_j(\mathbf{X})$, where $\sigma_j(\mathbf{X})$ is the $j$-th largest singular value of $\mathbf{X}$. The dual norm of the Schatten-1 norm is Schatten-$\infty$ norm (a.k.a., spectral norm) as $\|\mathbf{X}\|_{S_\infty} = \max_{j=1,\ldots,r} \sigma_j(\mathbf{X})$.

By Hölder's inequality, we have $|\langle \mathbf{W}, \mathbf{X} \rangle| \leq \|\mathbf{W}\|_{S_1} \|\mathbf{X}\|_{S_\infty}$. It is easy to prove a similar result for the overlapped Schatten-1 norm and its dual norm. We have the following Hölder-like inequality [22]:

$$|\langle \mathcal{W}, \mathcal{X} \rangle| \leq \|\mathcal{W}\|_{S_1} \|\mathcal{X}\|_{S_1^*} \leq \|\mathcal{W}\|_{S_1} \|\mathcal{X}\|_{\mathrm{mean}}, \tag{3}$$

where $\|\mathcal{X}\|_{\mathrm{mean}} := \frac{1}{K} \sum_{k=1}^{K} \|\mathbf{X}_{(k)}\|_{S_\infty}$.

Moreover, we define $\ell_1$-norm and $\ell_\infty$-norm for tensors that $\|\mathcal{X}\|_1 = \sum_{i_1=1}^{n_1} \ldots \sum_{i_K=1}^{n_K} |\mathcal{X}_{i_1,\ldots,i_K}|$, $\|\mathcal{X}\|_\infty = \max_{1 \leq i_1 \leq n_1} \ldots \max_{1 \leq i_K \leq n_K} |\mathcal{X}_{i_1,\ldots,i_K}|$. By Hölder's inequality, we have $|\langle \mathcal{W}, \mathcal{X} \rangle| \leq \|\mathcal{W}\|_1 \|\mathcal{X}\|_\infty$, and the following inequality relates the overlapped Schatten-1 norm with the Frobenius norm,

$$\|\mathcal{X}\|_{S_1} \leq \sum_{k=1}^{K} \sqrt{r_k} \|\mathcal{X}\|_F. \tag{4}$$

Let $\mathcal{W}^* \in \mathbb{R}^{n_1 \times \ldots \times n_K}$ be the low-rank tensor that we wish to recover. We assume that $\mathcal{W}^*$ is of rank $(r_1, \ldots, r_K)$. Thus, for each $k$, we have $\mathbf{W}^*_{(k)} = \mathbf{U}_k \mathbf{S}_k \mathbf{V}_k^\top$, where $\mathbf{U}_k \in \mathbb{R}^{n_k \times r_k}$ and $\mathbf{V}_k \in \mathbb{R}^{r_k \times n_k}$ are orthogonal matrices, which consist of left and right singular vectors of $\mathbf{W}^*_{(k)}$, $\mathbf{S}_k \in \mathbb{R}^{r_k \times r_k}$ is a diagonal matrix whose diagonal elements are singular values. Let $\Delta \in \mathbb{R}^{n_1 \times \ldots \times n_K}$

be an arbitrary tensor, we define the mode-$k$ orthogonal complement $\Delta_k''$ of its mode-$k$ unfolding $\Delta_{(k)} \in \mathbb{R}^{n_k \times N_{\setminus k}}$ with respect to the true low-rank tensor $\mathcal{W}^*$ as follows

$$\Delta_k'' = (\mathbf{I}_{n_k} - \mathbf{U}_k \mathbf{U}_k^\top) \Delta_{(k)} (\mathbf{I}_{\bar{N}_{\setminus k}} - \mathbf{V}_k \mathbf{V}_k^\top). \tag{5}$$

In addition $\Delta_k' = \Delta_{(k)} - \Delta_k''$ is the component which has overlapped row/column space with the unfolding of the true tensor $\mathbf{W}_{(k)}^*$. Note that the decomposition $\Delta_{(k)} = \Delta_k' + \Delta_k''$ is defined for each mode.

In [18], the concept of decomposibility and a large class of decomposable norms are discussed at length. Of particular relevance to us is the decomposability of the Schatten-1 norm and $\ell_1$-norm. We have the following equality, i.e., mode-$k$ decomposibility of the Schatten-1 norm that $\|\mathbf{W}_{(k)}^* + \Delta_k''\|_{S_1} = \|\mathbf{W}_{(k)}^*\|_{S_1} + \|\Delta_k''\|_{S_1}, k = 1, \ldots, K$. To note that the decomposibility is defined on each mode. It is also easy to check the decomposibility of the $\ell_1$-norm.

Let $\mathcal{V}^* \in \mathbb{R}^{n_1 \times \ldots \times n_K}$ be the gross corruption tensor that we wish to recover. We assume the the gross corruption is sparse, in that the cardinality $s = |\mathrm{supp}(\mathcal{V}^*)|$ of its support, $S = \mathrm{supp}(\mathcal{V}^*) = \{(i_1, i_2, \ldots, i_K) \in [n_1] \times \ldots \times [n_K] | \mathcal{V}_{i_1, \ldots, i_K}^* \neq 0\}$. This assumption leads to the inequality between the $\ell_1$ norm and the Forbenius norm that $\|\|\mathcal{V}^*\|\|_1 \leq \sqrt{s} \|\|\mathcal{V}^*\|\|_F$. Moreover, we have $\|\|\mathcal{V}^*\|\|_1 = \|\|\mathcal{V}_S^*\|\|_1$. For any $\mathcal{D} \in \mathbb{R}^{n_1 \times \ldots \times n_K}$, we have $\|\|\mathcal{D}\|\|_1 = \|\|\mathcal{D}_S\|\|_1 + \|\|\mathcal{D}_{S^c}\|\|_1$.

# 4 Main Results

To get a deep theoretical insight into the recovery property of robust tensor decomposition, we will now present a set of estimation error bounds. Unlike the analysis in [1], where only the summation of the estimation errors on the low-rank matrix and gross corruption matrix are analyzed, we aim at obtaining the estimation error bounds on each tensor (the low-rank tensor and corrupted tensor) separately. All the proofs can be found in the longer version of this paper.

Instead of considering the observation model in 1, we consider the following more general observation model

$$y_i = \langle \mathcal{W}^*, \mathcal{X}_i \rangle + \langle \mathcal{V}^*, \mathcal{X}_i \rangle + \epsilon_i, i = 1, \ldots, M, \tag{6}$$

where $\mathcal{X}_i$ can be seen as an observation operator, and $\epsilon_i$'s are i.i.d. zero mean Gaussian noise with variance $\sigma^2$. Our goal is to estimate an unknown rank $(r_1, \ldots, r_k)$ of tensor $\mathcal{W}^* \in \mathbb{R}^{n_1 \times \ldots \times n_K}$, as well as the unknown support of tensor $\mathcal{V}^*$, from observations $y_i, i = 1, \ldots, M$. We propose the following convex minimization to estimate the unknown low-rank tensor $\mathcal{W}^*$ and the sparse corruption tensor $\mathcal{V}^*$ simultaneously, with composite regularizers on $\mathcal{W}$ and $\mathcal{V}$ as follows:

$$(\widehat{\mathcal{W}}, \widehat{\mathcal{V}}) = \arg\min_{\mathcal{W}, \mathcal{V}} \frac{1}{2M} \|\mathbf{y} - \mathfrak{X}(\mathcal{W} + \mathcal{V})\|_2^2 + \lambda_M \|\|\mathcal{W}\|\|_{S_1} + \mu_M \|\|\mathcal{V}\|\|_1, \tag{7}$$

where $\mathbf{y} = (y_1, \ldots, y_M)^\top$ is the collection of observations, $\mathfrak{X}(\mathcal{W})$ is the linear observation model that $\mathfrak{X}(\mathcal{W}) = [\langle \mathcal{W}, \mathcal{X}_1 \rangle, \ldots, \langle \mathcal{W}, \mathcal{X}_M \rangle]^\top$. Note that (2) is a special case of (7), where the linear operator the identity tensor, we have $y_i$ as observation of each element in the summation of tensors $\mathcal{W}^* + \mathcal{V}^*$.

We also define $\mathbf{y}^* = (y_1^*, \ldots, y_M^*)^\top$, where $y_i^* = \langle \mathcal{W}^* + \mathcal{V}^*, \mathcal{X}_i \rangle$, is the true evaluation. Due to the noise of observation model, we have $\mathbf{y} = \mathbf{y}^* + \boldsymbol{\epsilon}$. In addition, we define the adjoint operator of $\mathfrak{X}$ as $\mathfrak{X}^* : \mathbb{R}^M \to \mathbb{R}^{n_1 \times \ldots \times n_K}$ that $\mathfrak{X}^*(\boldsymbol{\epsilon}) = \sum_{i=1}^M \epsilon_i \mathcal{X}_i$.

## 4.1 Deterministic Bounds

This section is devoted to obtain the deterministic bound of the residual low-rank tensor $\Delta = \widehat{\mathcal{W}} - \mathcal{W}^*$ and residual corruption tensor $\mathcal{D} = \widehat{\mathcal{V}} - \mathcal{V}^*$ separately, which makes our analysis unique.

We begin with a key technical lemma on residual tensors $\Delta = \widehat{\mathcal{W}} - \mathcal{W}^*$ and $\mathcal{D} = \widehat{\mathcal{V}} - \mathcal{V}^*$, obtained from the convex problem in (7).

**Lemma 1.** *Let $\widehat{\mathcal{W}}$ and $\widehat{\mathcal{V}}$ be the solution of minimization problem* (7) *with $\lambda_M \geq 2 \|\|\mathfrak{X}^*(\boldsymbol{\epsilon})\|\|_{\mathrm{mean}}/M$, $\mu_M \geq 2 \|\|\mathfrak{X}^*(\boldsymbol{\epsilon})\|\|_\infty/M$, we have*

1. $\mathrm{rank}(\boldsymbol{\Delta}'_k) \le 2r_k$.

2. *There exist $\beta_1 \ge 3$ and $\beta_2 \ge 3$, such that $\sum_{k=1}^{K} \|\boldsymbol{\Delta}''_k\|_{S_1} \le \beta_1 \sum_{k=1}^{K} \|\boldsymbol{\Delta}'_k\|_{S_1}$ and $\||\mathcal{D}_{S^c}|\|_1 \le \boldsymbol{\beta_2} \||\mathcal{D}_S|\|_1$.*

The lemma can be obtained by utilizing the optimality of $\widehat{\mathcal{W}}$ and $\widehat{\mathcal{V}}$, as well as the decomposibility of Schatten-1 norm and $\ell_1$-norm of tensors.

Also, we obtain the key property of the optimal solution of (7), presented in the following theorem.

**Theorem 1.** *Let $\widehat{\mathcal{W}}$ and $\widehat{\mathcal{V}}$ be the solution of minimization problem (7) with $\lambda_M \ge 2 \||\mathfrak{X}^*(\boldsymbol{\epsilon})|\|_{\mathrm{mean}}/M$, $\mu_M \ge 2 \||\mathfrak{X}^*(\boldsymbol{\epsilon})|\|_\infty/M$, we have*

$$\frac{1}{2M}\|\mathfrak{X}(\Delta + \mathcal{D})\|_2^2 \le \frac{3\lambda_M}{2K}\sum_{k=1}^{K}\|\boldsymbol{\Delta}'_k\|_{S_1} + \frac{3\mu_M}{2}\||\mathcal{D}_S|\|_1. \tag{8}$$

Theorem 1 provides a deterministic prediction error bound for model (7). This is a very general result, and can be applied to any linear operator $\mathfrak{X}$, including the robust tensor decomposition case that we are particularly interested in this paper. It also covers, for example, tensor regression, tensor compressive sensing, to mention a few.

Furthermore, we impose an assumption on the linear operator and the residual low-rank tensor and residue sparse corruption tensor, which generalized the restricted eigenvalue assumption [2] [10].

**Assumption 1.** *Defining $\Omega = \{(\Delta, \mathcal{D})| \sum_{k=1}^{K} \|\boldsymbol{\Delta}''_k\|_{S_1} \le \beta_1 \sum_{k=1}^{K} \|\boldsymbol{\Delta}'_k\|_{S_1}, \||\mathcal{D}_{S^c}|\|_1 \le \beta_2 \||\mathcal{D}_S|\|_1\}$, we assume there exist positive scalars $\kappa_1$, $\kappa_2$ that*

$$\kappa_1 = \min_{\Delta, \mathcal{D} \in \Omega} \frac{\|\mathfrak{X}(\Delta + \mathcal{D})\|_2}{\sqrt{M} \||\Delta|\|_F} > 0, \quad \kappa_2 = \min_{\Delta, \mathcal{D} \in \Omega} \frac{\|\mathfrak{X}(\Delta + \mathcal{D})\|_2}{\sqrt{M} \||\mathcal{D}|\|_F} > 0.$$

Note that Assumption 1 is also related to restricted strong convexity assumption, which is proposed in [18] to analyze the statistical properties of general M-estimators in the high dimensional setting.

Combing the results in Theorem 1 and Assumption 1, we have the following theorem, which summarizes our main result.

**Theorem 2.** *Let $\widehat{\mathcal{W}}, \widehat{\mathcal{V}}$ be an optimal solution of (7), and take the regularization parameters $\lambda_M \ge 2 \||\mathfrak{X}^*(\boldsymbol{\epsilon})|\|_{\mathrm{mean}}/M$, $\mu_M \ge 2 \||\mathfrak{X}^*(\boldsymbol{\epsilon})|\|_\infty/M$. Then the following results hold:*

$$\left\||\widehat{\mathcal{W}} - \mathcal{W}^*|\right\|_F \le \frac{3}{\kappa_1}\left(\frac{1}{K}\sum_{k=1}^{K}\frac{\lambda_M\sqrt{2r_k}}{\kappa_1} + \frac{\mu_M\sqrt{s}}{\kappa_2}\right), \tag{9}$$

$$\left\||\widehat{\mathcal{V}} - \mathcal{V}^*|\right\|_F \le \frac{3}{\kappa_2}\left(\frac{1}{K}\sum_{k=1}^{K}\frac{\lambda_M\sqrt{2r_k}}{\kappa_1} + \frac{\mu_M\sqrt{s}}{\kappa_2}\right). \tag{10}$$

Theorem 2 provides us with the error bounds of each tensor separately. Specifically, these bounds not only measure how well our decomposition model can approximate the observation model defined in (6), but also measure how well the decomposition of the true low-rank tensor and gross corruption tensor is. When $s = 0$, our theoretical results reduce to that proposed in [22], which is a special case of our problem, i.e., noisy low-rank tensor decomposition without corruption.

On the other hand, the results obtained in Theorem 2 are very appealing both practically and theoretically. From the perspective of applications, this result is quite useful as it helps us to better understand the behavior of each tensor separately. From the theoretical point of view, this result is novel, and is incomparable with previous results [1][17] or simple generalization of previous results.

Though Theorem 2 has provided estimation error bounds of $\widehat{\mathcal{W}}$ and $\widehat{\mathcal{V}}$, it is unclear whether the rank of $\mathcal{W}^*$ and the support of $\mathcal{V}^*$ can be exactly recovered. We show that under some assumptions about the true tensors, both of them can be exactly recovered.

**Corollary 1.** *Under the same conditions of Theorem 2, if the following condition holds:*

$$\sigma_{r_k}(\mathbf{W}^*_{(k)}) > \frac{6(1 + \beta_1)\sum_{k=1}^{K}\sqrt{2r_k}}{\kappa_1 MK}\left(\frac{1}{K}\sum_{k=1}^{K}\frac{\lambda_M\sqrt{2r_k}}{\kappa_1} + \frac{\mu_M\sqrt{s}}{\kappa_2}\right), \tag{11}$$

*where $\sigma_{r_k}(\mathbf{W}^*_{(k)})$ is the $r_k$-th largest singular value of $\mathbf{W}^*_{(k)}$, then*

$$\widehat{r}_k = \left\{ \arg\max_r \sigma_r(\widehat{\mathbf{W}}_{(k)}) > \frac{3(1+\beta_1)\sum_{k=1}^K \sqrt{2r_k}}{\kappa_1 MK} \left( \frac{1}{K} \sum_{k=1}^K \frac{\lambda_M \sqrt{2r_k}}{\kappa_1} + \frac{\mu_M \sqrt{s}}{\kappa_2} \right) \right\}$$

*recovers the rank of $\mathbf{W}^*_{(k)}$ for all $k$.*

*Furthermore, if the following condition holds:*

$$\min_{i_1,\ldots,i_K} |\mathcal{V}^*_{i_1,\ldots,i_K}| > \frac{6(1+\beta_2)\sqrt{s}}{\kappa_2 M} \left( \frac{1}{K} \sum_{k=1}^K \frac{\lambda_M \sqrt{2r_k}}{\kappa_1} + \frac{\mu_M \sqrt{s}}{\kappa_2} \right), \tag{12}$$

*then*

$$\widehat{S} = \left\{ (i_1, i_2, \ldots, i_K) : \widehat{\mathcal{V}}_{i_1,\ldots,i_K} > \frac{3(1+\beta_2)\sqrt{s}}{\kappa_2 M} \left( \frac{1}{K} \sum_{k=1}^K \frac{\lambda_M \sqrt{2r_k}}{\kappa_1} + \frac{\mu_M \sqrt{s}}{\kappa_2} \right) \right\}$$

*recovers the true support of $\mathcal{V}^*$.*

Corollary 1, basically states that, under the assumption that the singular values of the low-rank tensor $\mathcal{W}^*$, and the entry values of corruption tensor $\mathcal{V}^*$ are above the noise level (e.g., (11) and (12)), we can recover the rank and the support successfully.

## 4.2  Noisy Tensor Decomposition

Now we are going back to study robust tensor decomposition with corruption in (2), which is a special case of (7), where the linear operator is identity tensor. As the linear operator $\mathfrak{X}$ is a vectorization such that $M = N$, and $\|\mathfrak{X}(\Delta + \mathcal{D})\|_2 = \|\|\Delta + \mathcal{D}\|\|_F$. In addition, it is easy to show that Assumption 1 holds with $\kappa_1 = \kappa_2 = O(1/\sqrt{N})$. It remains to bound $\|\|\mathfrak{X}^*(\boldsymbol{\epsilon})\|\|_{\text{mean}}$ and $\|\|\mathfrak{X}^*(\boldsymbol{\epsilon})\|\|_\infty$, as shown in the following lemma [1] [24].

**Lemma 2.** *Suppose that $\mathfrak{X} : \mathbb{R}^{n_1 \times \cdots \times n_K} \to \mathbb{R}^N$ is a vectorization of a tensor. Then we have with probability at least $1 - 2\exp(-C(n_k + \bar{N}_{\backslash k})) - 1/N$ that*

$$\|\|\mathfrak{X}^*(\boldsymbol{\epsilon})\|\|_{\text{mean}} \le \frac{\sigma}{K} \sum_{k=1}^K \left( \sqrt{n_k} + \sqrt{\bar{N}_{\backslash k}} \right),$$

$$\|\|\mathfrak{X}^*(\boldsymbol{\epsilon})\|\|_\infty \le 4\sigma\sqrt{\log N},$$

*where $C$ is a universal constant.*

With Theorem 2 and Lemma 2, we immediately have the following estimation error bounds for robust tensor decomposition.

**Theorem 3.** *Suppose that $\mathfrak{X} : \mathbb{R}^{n_1 \times \cdots \times n_K} \to \mathbb{R}^N$ is a vectorization of a tensor. Then for the regularization constants $\lambda_N \ge 2\sigma \sum_{k=1}^K \left( \sqrt{n_k} + \sqrt{\bar{N}_{\backslash k}} \right)/(NK)$, $\mu_N > 8\sigma\sqrt{\log N}/N$, with probability at least $1 - 2\exp(-C(n_k + \bar{N}_{\backslash k})) - 1/N$, any solution of (2) have the following error bound:*

$$\left\|\left\|\widehat{\mathcal{W}} - \mathcal{W}^*\right\|\right\|_F \le \frac{6}{\kappa_1} \left( \frac{1}{K} \sum_{k=1}^K \frac{\sigma \sum_{k=1}^K \left( \sqrt{n_k} + \sqrt{\bar{N}_{\backslash k}} \right) \sqrt{2r_k}}{\kappa_1 NK} + \frac{4\sigma\sqrt{s\log N}}{\kappa_2 N} \right),$$

$$\left\|\left\|\widehat{\mathcal{V}} - \mathcal{V}^*\right\|\right\|_F \le \frac{6}{\kappa_2} \left( \frac{1}{K} \sum_{k=1}^K \frac{\sigma \sum_{k=1}^K \left( \sqrt{n_k} + \sqrt{\bar{N}_{\backslash k}} \right) \sqrt{2r_k}}{\kappa_1 NK} + \frac{4\sigma\sqrt{s\log N}}{\kappa_2 N} \right).$$

In the special case that $n_1 = \ldots = n_K = n$ and $r_1 = \ldots = r_K = r$, we have $\left\|\left\|\widehat{\mathcal{W}} - \mathcal{W}^*\right\|\right\|_F = O(\sigma\sqrt{rn^{K-1}} + \sigma\sqrt{Ks\log n})$ and $\left\|\left\|\widehat{\mathcal{V}} - \mathcal{V}^*\right\|\right\|_F = O(\sigma\sqrt{rn^{K-1}} + \sigma\sqrt{Ks\log n})$, which matches the error bound of robust matrix decomposition [1] when $K = 2$.

Note that the high probability support and rank recovery guarantee for the special case of tensor decomposition follows immediately from Corollary 1. Due to the space limit, we omit the result here.

# 5 Algorithm

In this section, we present an algorithm to solve (2). Since (2) is a special case of (7), we consider the more general problem (7). It is easy to show that (7) is equivalent to the following problem with auxiliary variables $\Psi, \Phi$:

$$\min_{\mathcal{W}, \mathcal{V}, \mathcal{Y}, \mathcal{Z}} \frac{1}{2M} \|\mathbf{y} - \mathbf{x}^\top (\mathbf{w} + \mathbf{v})\|_2^2 + \frac{\lambda_M}{K} \sum_{k=1}^{K} \||\Psi_k\||_{S_1} + \frac{\mu_M}{K} \sum_{k=1}^{K} \||\Phi_k\||_1,$$

$$\text{subject to } \mathbf{P}_k \mathbf{w} = \boldsymbol{\psi}_k, \mathbf{P}_k \mathbf{v} = \boldsymbol{\phi}_k,$$

where $\mathbf{x}, \mathbf{w}, \mathbf{v}, \boldsymbol{\psi}_k, \boldsymbol{\phi}_k$ are the vectorizations of $\sum_{i=1}^{M} \mathcal{X}_i, \mathcal{W}, \mathcal{V}, \Psi_k, \Phi_k$ respectively, and $\mathbf{P}_k$ is the transformation matrix that change the order of rows and columns so that $\mathbf{P}_k \mathbf{w} = \boldsymbol{\psi}_k$.

The augmented Lagrangian (AL) function of the above minimization problem with respect to the primal variables $(\mathcal{W}^t, \mathcal{V}^t)$ is given as follows:

$$L_\eta(\mathcal{W}, \mathcal{V}, \{\Psi_k\}_{k=1}^K, \{\Phi_k\}_{k=1}^K, \{\boldsymbol{\alpha}_k\}_{k=1}^K, \{\boldsymbol{\beta}_k\}_{k=1}^K)$$

$$= \frac{1}{2} \|\mathbf{y} - \mathbf{x}^\top (\mathbf{w} + \mathbf{v})\|_2^2 + \frac{\lambda_M M}{K} \sum_{k=1}^{K} \||\Psi_k\||_{S_1} + \frac{\mu_M M}{K} \sum_{k=1}^{K} \||\Phi_k\||_1$$

$$+ \eta \left( \sum_k (\boldsymbol{\alpha}_k^\top (\mathbf{P}_k \mathbf{w} - \boldsymbol{\psi}_k) + \frac{1}{2} \|\mathbf{P}_k \mathbf{w} - \boldsymbol{\psi}_k\|_2^2) + \sum_k (\boldsymbol{\beta}_k^\top (\mathbf{P}_k \mathbf{v} - \boldsymbol{\phi}_k) + \frac{1}{2} \|\mathbf{P}_k \mathbf{v} - \boldsymbol{\phi}_k\|_2^2) \right),$$

where $\boldsymbol{\alpha}^t, \boldsymbol{\beta}^t$ are Lagrangian multiplier vectors, and $\eta > 0$ is a penalty parameter.

We then apply the algorithm of Alternating Direction Method of Multipliers (ADMM) [3, 20] to solve the above optimization problem. Starting from initial points $(\mathbf{w}^0, \mathbf{v}^0, \{\Psi_k^0\}_{k=1}^K, \{\Phi_k^0\}_{k=1}^K, \{\boldsymbol{\alpha}_k^0\}_{k=1}^K, \{\boldsymbol{\beta}_k^0\}_{k=1}^K)$, ADMM performs the following updates iteratively:

$$\mathbf{w}^{t+1} = \left( (\mathbf{x}^\top \mathbf{y} - \mathbf{x}^\top \mathbf{x} \mathbf{v}^t) + \eta \sum_{k=1}^{K} \mathbf{P}_k^\top (\boldsymbol{\psi}_k^t - \boldsymbol{\alpha}_k^t) \right) / (1 + \eta K),$$

$$\mathbf{v}^{t+1} = \left( (\mathbf{x}^\top \mathbf{y} - \mathbf{x}^\top \mathbf{x} \mathbf{w}^{t+1}) + \eta \sum_{k=1}^{K} \mathbf{P}_k^\top (\boldsymbol{\phi}_k^t - \boldsymbol{\beta}_k^t) \right) / (1 + \eta K),$$

$$\Psi_k^{t+1} = \text{prox}_{\frac{\lambda M}{\eta K}}^{tr} (\mathbf{P}_k \mathbf{w}^{t+1} + \boldsymbol{\alpha}_k^t), \qquad \Phi_k^{t+1} = \text{prox}_{\frac{\mu M}{\eta K}}^{\ell_1} (\mathbf{P}_k \mathbf{v}^{t+1} + \boldsymbol{\beta}_k^t) \qquad k = 1, \dots, K,$$

$$\boldsymbol{\alpha}_k^{t+1} = \boldsymbol{\alpha}_k^{t+1} + (\mathbf{P}_k \mathbf{w}^{t+1} - \boldsymbol{\psi}_k^{t+1}) \qquad \boldsymbol{\beta}_k^{t+1} = \boldsymbol{\beta}_k^{t+1} + (\mathbf{P}_k \mathbf{v}^{t+1} - \boldsymbol{\phi}_k^{t+1}) \qquad k = 1, \dots, K,$$

where $\text{prox}_\gamma^{tr}(\cdot)$ is the soft-thresholding operator for trace norm, and $\text{prox}_\gamma^{\ell_1}(\cdot)$ is the soft-thresholding operator for $\ell_1$ norm [4, 11]. The stopping criterion is that all the partial (sub)gradients are (near) zero, under which condition we obtain the saddle point of the augmented Lagrangian function. Since (7) is strictly convex, the saddle point is the global optima for the primal problem.

# 6 Experiments

In this section, we conduct numerical experiments to confirm our analysis in previous sections. The experiments are conducted under the setting of robust noisy tensor decomposition.

We follow the procedure described in [22] for the experimental part. We randomly generate low-rank tensors of dimensions $\boldsymbol{n}^{(1)} = (50, 50, 20)$ ( results are shown in Figure 1(a, b, c)) and $\boldsymbol{n}^{(2)} = (100, 100, 50)$( results are shown in Figure 1(d, e, f)) for various rank $(r_1, r_2, ..., r_k)$. Given a specific rank, we first generated the "core tensor" with elements $r_1 \times \ldots \times r_K$ from the standard normal distribution, and then multiplied each mode of the core tensor with an orthonormal factor randomly drawn from the Haar measure. For the gross corruption, we randomly generated the sparsity of the corruption matrix $s$, and then randomly selected $s$ elements in which we put values randomly generated from uniform distribution. The additive independent Gaussian noise with variance $\sigma^2$

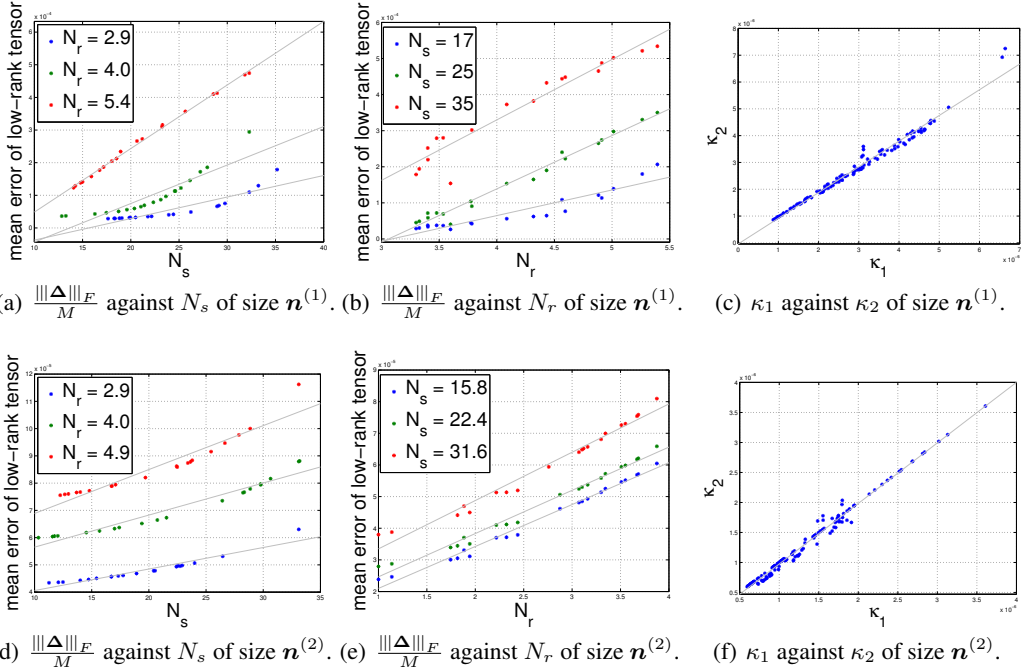

(a) $\frac{\|\|\mathbf{\Delta}\|\|_F}{M}$ against $N_s$ of size $\boldsymbol{n}^{(1)}$. (b) $\frac{\|\|\mathbf{\Delta}\|\|_F}{M}$ against $N_r$ of size $\boldsymbol{n}^{(1)}$. (c) $\kappa_1$ against $\kappa_2$ of size $\boldsymbol{n}^{(1)}$.

(d) $\frac{\|\|\mathbf{\Delta}\|\|_F}{M}$ against $N_s$ of size $\boldsymbol{n}^{(2)}$. (e) $\frac{\|\|\mathbf{\Delta}\|\|_F}{M}$ against $N_r$ of size $\boldsymbol{n}^{(2)}$. (f) $\kappa_1$ against $\kappa_2$ of size $\boldsymbol{n}^{(2)}$.

Figure 1: Results of robust noisy tensor decomposition with corruption, under different sizes.

was added to the observations of elements. We use the alternating direction method of multipliers (ADMM) to solve the minimization problem (2). The whole experiments were repeated 50 times and the averaged results are reported.

The results are shown in Figure 1, where $N_r = \sum_{k=1}^{K} \sqrt{r_k}/K$, and $N_s = \sqrt{s}$. In Figure 1(a, d), we first fix $N_r$ at different values, and then draw the value of $\|\|\widehat{\mathcal{W}} - \mathcal{W}^*\|\|_F /N$ against $N_s$. Similarly, in Figure 1(b, e), we first fix $N_s$ at different values, and then draw $\|\|\widehat{\mathcal{W}} - \mathcal{W}^*\|\|_F /N$ against $N_r$. In Figure 1(c, f), we study the values of $\kappa_1$ and $\kappa_2$ at various settings. We can see that $\|\|\widehat{\mathcal{W}} - \mathcal{W}^*\|\|_F /N$ scales linearly with both $N_s$ and $N_r$. Similar scalings of $\|\|\widehat{\mathcal{V}} - \mathcal{V}^*\|\|_F /N$ can be observed, hence we omit them due to space limitation. We can also observe from Figure 1(c, f) that, under various settings, $\kappa_1 \approx \kappa_2$, this finding is consistent with the fact that $\|\|\widehat{\mathcal{W}} - \mathcal{W}^*\|\|_F /N \approx \|\|\widehat{\mathcal{V}} - \mathcal{V}^*\|\|_F /N$. All these results are consistent with each other, validating our theoretical analysis.

## 7   Conclusions

In this paper, we analyzed the statistical performance of robust noisy tensor decomposition with corruption. Our goal is to recover a pair of tensors, based on observing a noisy contaminated version of their sum. It is based on solving a convex optimization with composite regularizations of Schatten-1 norm and $\ell_1$ norm defined on tensors. We provided a general nonasymptotic estimator error bounds on the underly low-rank tensor and sparse corruption tensor. Furthermore, the error bound we obtained in this paper is new, and non-comparable with previous theoretical analysis.

## Acknowledgement

We would like to thank the anonymous reviewers for their helpful comments. Research was sponsored in part by the Army Research Lab, under Cooperative Agreement No. W911NF-09-2-0053 (NSCTA), the Army Research Office under Cooperative Agreement No. W911NF-13-1-0193, National Science Foundation IIS-1017362, IIS-1320617, and IIS-1354329, HDTRA1-10-1-0120, and MIAS, a DHS-IDS Center for Multimodal Information Access and Synthesis at UIUC.

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
