[Reviews · NeurIPS 2014]

Submitted by Assigned_Reviewer_1

The paper considers the problem of recovering a low-rank tensor and sparse corruption from noisy linear measurements of the sum. This is done by a convex program that uses overlapped Schatten-1 norm penalty on the low rank component (which is sum of singular values of unfolded tensors along each mode) and a L1 penalty on the sparse component.
The main result of the paper is individual estimation error bounds for low-rank and sparse components in term of Frobenius norm. The paper also presents a ADMM based algorithm for optimizing the objective.

The estimation error for sparse component in Eq 10 is in terms of Frobenius norm which is less informative for sparse component -- support recovery guarantees would be more insightful.

Summary: The paper is well-written and make a clear contribution to robust low-rank tensor decomposition.

Submitted by Assigned_Reviewer_3

The authors provide a novel algorithm for recovering a low-rank tensor from a set of noisy observations of the tensor. This work extends, in a non-trivial way, previous results obtained in the matrix case. The theoretical analysis of the algorithm in Section 4 is very appealing and the preliminary numerical experiments in Section 5 are encouraging.
Summary: The authors provide a novel algorithm for recovering a low-rank tensor from a set of noisy observations of the tensor. This work extends, in a non-trivial way, previous results obtained in the matrix case. The theoretical analysis of the algorithm in Section 4 is very appealing and the preliminary numerical experiments in Section 5 are encouraging.

Submitted by Assigned_Reviewer_28

This paper studies low rank tensor recovery problem with gross corruption using convex program. The convex programs considered here seem to be inspired by matrix completion and tensor trace norms. The main contribution is in handling gross corruption, which is a very natural noise model. The paper also involves some synthetic experiments.

Overall this is an interesting paper. The analysis is clean. The results are reasonable but the bounds involves many parameters and would be good if the authors can give more intuitions on different trade-offs. For example, although Assumption 1 is reasonable (e.g. it holds for random subspace \Omega and random sparsity pattern S as long as the rank/sparsity is not too large), it is not immediately clear and would be good to discuss some typical cases where it is true. It would also be good if the bound can be simplified (for example in the case where all the n_k's are the same, r_k's are also the same) to provide more intuition.

Another (slight) concern is that the authors are mixing tensor decomposition with tensor completion in the intro. Although these are closely related, tensor decomposition (especially CP decomposition) is often harder than tensor completion (also for low rank tensor completion typically the definition of rank is different from CP rank, as in this paper).
Summary: This paper studies low rank tensor recovery problem with gross corruption using convex program. The analysis is clean and interesting.
Author Feedback
Author rebuttal: Thank you for your valuable comments on our manuscript.

Response to Assigned_Reviewer_1

Thank you for your constructive comments.

Q1:“The estimation error for sparse component in Eq 10 is in terms of Frobenius norm which is less informative for sparse component -- support recovery guarantees would be more insightful.”

A1: Eq 10 provides the estimation error bound of the sparse component, which might be of interest. It is true that recovery guarantees is more insightful. In fact, we do provide the support recovery guarantees for the sparse component in Corollary 2 (Line 283). Note that Corollary 2 is for the general formulation in Eq 7. Thus, the support recovery guarantee for the special case of tensor decomposition immediately follows from Corollary 2. We will emphasize it in the final version.

Response to Assigned_Reviewer_2

Thank you for your helpful comments.

Q1:“Assumption 1 is reasonable (e.g. it holds for random subspace \Omega and random sparsity pattern S as long as the rank/sparsity is not too large), it is not immediately clear and would be good to discuss some typical cases where it is true.”

A1: Thank you for your suggestion. Assumption 1 is an extension of restricted eigenvalue condition. This is a mild assumption. We will give some examples when Assumption 1 holds in the final version.

Q2:“It would also be good if the bound can be simplified (for example in the case where all the n_k's are the same, r_k's are also the same) to provide more intuition”

A2: To better understand the bounds, take Theorem 2 as an example, a very intuitive explanation is that the error bound is actually controlled by both the rank of the low-rank component and the sparsity of the sparse component. The higher the rank of the low-rank component, and the denser the sparse component, the larger estimation error we will obtain. We will add a discussion of the special case, where all n_k’s and r_k’s are the same in the final version. Thank you for your suggestion.

Q3: “Another (slight) concern is that the authors are mixing tensor decomposition with tensor completion in the intro.”

A3: We are sorry for the confusion about tensor decomposition and tensor completion in the introduction section. We are aware that the tensor decomposition problem is actually harder than the tensor completion problem. We will carefully revise the confusing presentation in the introduction. Thank you for pointing out this minor issue.

Response to Assigned_Reviewer_3

Thank you for your insightful and nice comments

In summary, we would like to thank the reviewers again for their helpful comments. All the issues pointed out by the reviewers are easy to address (or have been already addressed in the current submission, and just need to be highlighted and emphasized). We will incorporate them in the camera ready version.